**Subject Category:**
Biology (whole organism)

ecology/behaviour

insect migration, Syrphidae, hoverfly, Diptera, pollination, pest control

**Author for correspondence:**
Karl R. Wotton
e-mail: k.r.wotton@exeter.ac.uk

# Quantification of migrant hoverfly movements (Diptera: Syrphidae) on the West Coast of North America

Myles H. M. Menz[1,2,3], Brian V. Brown[4]
and Karl R. Wotton[5]

[1]Department for Migration and Immuno-Ecology, Max Planck Institute for Ornithology, 78315 Radolfzell, Germany
[2]Institute of Ecology and Evolution, University of Bern, 3012 Bern, Switzerland
[3]School of Biological Sciences, The University of Western Australia, Crawley 6009, Western Australia, Australia
[4]Entomology Section, Natural History Museum of Los Angeles County, Exposition Boulevard, Los Angeles, CA, USA
[5]Centre for Ecology and Conservation, College of Life and Environmental Sciences, University of Exeter, Penryn, Cornwall TR10 9FE, UK

MHMM, 0000-0002-3347-5411; KRW, 0000-0002-8672-9948

The seasonal migration of huge numbers of hoverflies is frequently reported in Europe from mountain passes or spurs of land. The movement of such large numbers of beneficial insects is thought to provide significant ecosystem services in terms of pollination and pest control. Observations from the East Coast of the USA during the 1920s indicate the presence of migratory life histories among some hoverfly species there, but 90 years have now passed since the last reported observation of hoverfly migration in the USA. Here, we analyse video footage taken during a huge northward migration of hoverflies on 20 April 2017 on the West Coast of California. The quantification of migrant numbers from this footage allows us to estimate the passage of over 100 000 hoverflies in half an hour over a 200 m section of headland in Montaña de Oro State Park (San Luis Obispo County). Field collections and analysis of citizen science data indicate different species from the previously reported *Eristalis tenax* migrations on the East Coast of the USA and provide evidence for migration among North American hoverflies. We wish to raise awareness of this phenomenon and suggest approaches to advance the study of hoverfly migration in North America and elsewhere.

# 1. Introduction

Insect migration is an environmentally significant phenomenon that provides both advantages and challenges to human populations. Worldwide, there are many migratory insect species but only a handful of iconic migrants, mostly made up of large butterflies, dragonflies or major crop pests [1,2]. While the migratory journeys of these insects attract considerable attention, other less conspicuous groups such as hoverflies (sometimes referred to as flower flies in North America; Diptera: Syrphidae) remain poorly studied, despite their important role in pollination and biological control of crop pests such as aphids [3–5]. The migration of hoverflies is best understood in Europe where seasonal influxes into northern regions begin around May and are followed by often-huge southerly migrations during August–October. These migrations take place on a broad front and, in general, observations are restricted to locations that concentrate large numbers of migrants—spurs of land bordered by water bodies [6,7] or mountain passes [8–10].

Reports of hoverfly migration in other regions of the world, although sparse, exist from Nepal [11], the East Coast of the USA [12] and perhaps Australia [13], suggesting that the phenomenon is widespread, if poorly documented. Large autumn migrations on the US East Coast of one hoverfly species, the drone fly *Eristalis tenax* (also a European migrant and cosmopolitan species; [8,10]), were reported numerous times between 1915 and 1926 as they followed the coast south [12]. Surprisingly, we have found no reports of this phenomenon in *E. tenax*—or in any other hoverfly—in the more than 90 years that have passed since, and the current status of hoverfly migration in the USA appears unknown. However, on 20 April 2017, BVB witnessed a large migration of hoverflies, this time on the West Coast of California in San Luis Obispo County (figure 1). We analysed this hitherto unknown migration using a combination of video data obtained during the migration, captures of hoverflies from San Luis Obispo County and analysis of citizen science data. Our results reveal the huge scale of this migration event and identify different migratory species to those observed previously on the East Coast.

# 2. Material and methods

## 2.1. Image analysis

Migrant numbers were quantified from a video recording taken at 30 fps and 8 MP resolution with an Olympus E-M1 mark II camera (electronic supplementary material, file S1). The video was converted to an uncompressed AVI file using FFMPEG software (https://www.ffmpeg.org) and each frame visualized with ImageJ (https://imagej.nih.gov/ij/). Hoverfly numbers were quantified as they passed a set point a third of the way through the field of view. High-resolution 20 MP still images of the hoverfly migration were analysed to quantify numbers and identify potential species.

## 2.2. Candidate species identification

Hoverflies were collected from San Luis Obispo County (California) over a two-week period in April 2018 using hand nets. Permission was obtained from the State of California Natural Resources Agency: Department of Parks and Recreation (Ref: 17-820-38). Distributional data for the candidate species were obtained from the citizen science website iNaturalist (https://www.inaturalist.org). We searched for verifiable records using the description 'Syrphidae' covering North America. Subsequently, we selected data west of $-114°$ longitude to represent the West Coast. Raw data are available in the electronic supplementary material, file S4.

# 3. Results

Our observations of hoverfly migration took place on 20 April 2017 from 10.00 to 10.30 at an elevation of $80-100$ m above the nearby sea level at Montaña de Oro State Park, California (figure 1). Many thousands of flies appeared to be present, all moving northward against a brisk headwind and not stopping on nearby flowers or vegetation (figure 2*a*). We were able to estimate hoverfly numbers based on a video recording documenting the migration (electronic supplementary material, file S1). We detected the transit of 660 hoverflies during the 35 s recording. When scaled to the 30 min of observed migration, this is around 34 000 hoverflies passing over this small section of trail no wider

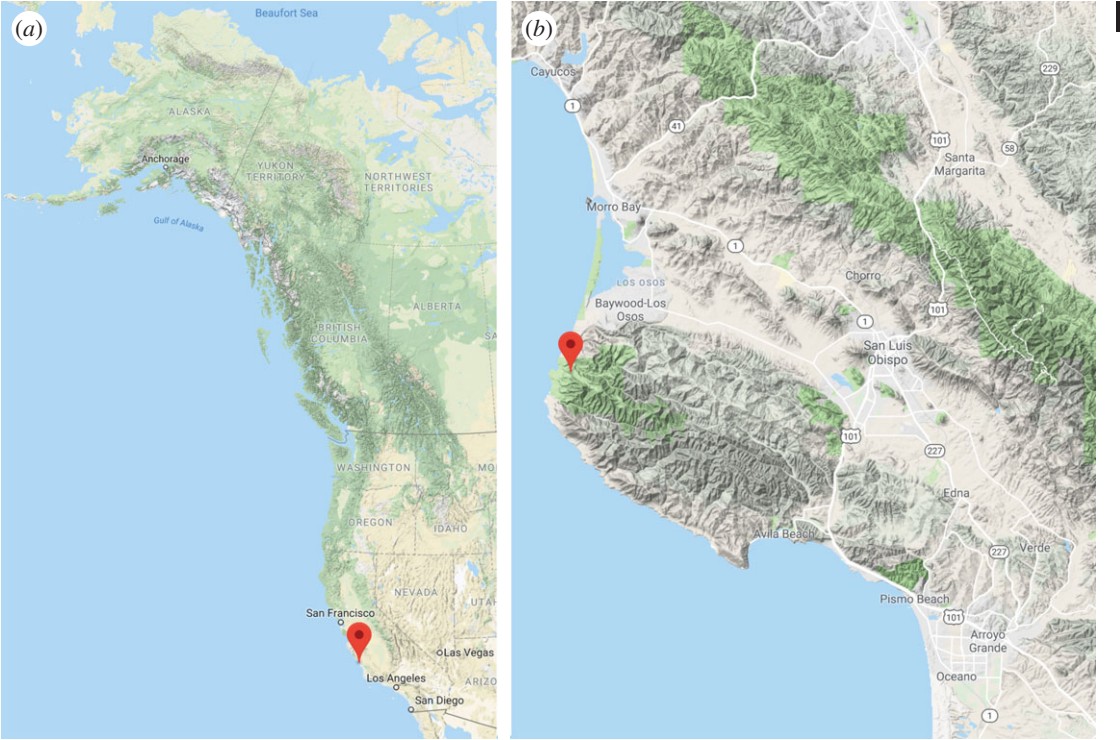

**Figure 1.** Location of hoverfly migration on the West Coast of North America. (*a*) Location of hoverfly migration observed on 20 April 2017 at 35.27° N, 120.88° W (red pin) and the West Coast regions investigated in this study. (*b*) Detailed view of the observation site on the Valencia Peak trail in Montaña de Oro State Park, San Luis Obispo County, California, USA, and the coastal topology of this region. Images taken from Google Maps.

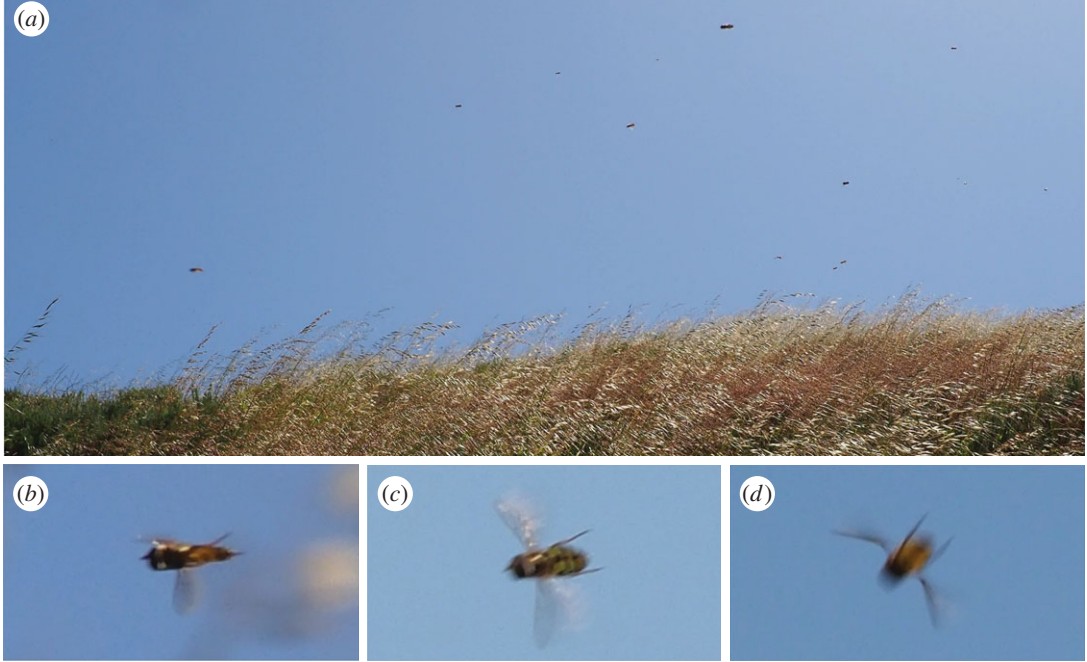

**Figure 2.** (*a*) Image of hoverfly migration observed on the Valencia Peak trail in Montaña de Oro State Park. This image is a representative frame extracted from a video recording; 19 individual hoverflies can be seen migrating to the north (right to left) against the background of the sky. (*b*–*d*) Images of migrating hoverflies in flight: (*b*) frontolateral view; (*c*) dorsal view; (*d*) ventral view with dark ventral bars visible.

than 20 m. Analysis of higher resolution still images taken from the same position as the recording, and without quality reduction brought about by video compression, allowed us to detect four times the number of migrants in each image (electronic supplementary material, files S2 and S3). These numbers indicate around 136 000 hoverflies may have passed this point; however, migration was observed at a similar density over an estimated 200 m wide section of trail, indicating that total numbers may have considerably exceeded this.

Unfortunately, no specimens were captured during this observation, but high-resolution photos taken at random were used to identify the species involved (figure 2*b*–*d*). Analysis of these images suggests a single species of relatively large, predominantly black and yellow/orange hoverfly. This coloration and appearance allow us to exclude *E. tenax*, which was previously observed to migrate along the East Coast of the USA by Shannon [12]. To detect candidate species, we captured a range of hoverflies from San Luis Obispo County during the spring of 2018 and identified four candidates, whose congeners are known to be migrants in Europe [8,10], and that closely resemble the hoverflies in our observations. These are the black and white *Scaeva pyrastri* (also a migrant in Europe), the black and yellow *Syrphus opinator*, and the smaller *Eupodes fumipennis* and *Eupodes volucris*.

To identify if these candidate species also had ranges that extended up the West Coast through spring and summer, indicative of migratory movements, we obtained records from the iNaturalist database (https://www.inaturalist.org). Records from *Syrphus*, *Scaeva* and *Eupeodes* spp. all occurred in high abundance in California in April, with observations occurring up through Oregon, Washington, British Columbia and into Alaska as spring commenced into summer, then through increasingly lower latitudes through the autumn and winter (figure 3). Next, we asked if any of these species was more abundant in California during 2017 than in other years. Of our candidates, only *S. pyrastri* was more abundant in 2017 ($n = 42$ versus 23 in 2016 and 2018).

## 4. Discussion

Northward movements of migratory insects are known to occur throughout North America during the spring in response to seasonal changes [1], yet over 90 years have now passed since the last reports of hoverfly migration in the USA. Our results indicate a clear adaptive movement of hoverflies to the north in the spring as is seen for other migratory insects, a conclusion strengthened by their active flight against the prevailing wind. Our results also demonstrate the scale of hoverfly movement, which we estimate reached a minimum of 34 000 flies in the 30 min of observation but probably totalled in the hundreds of thousands over the complete migration front. It remains unclear, however, for what length of time this event occurred. In Europe, similar events occur over a number of days during April, May and June, during which a multi-generational movement to the north occurs; hence, a similar pattern in North America would indicate the involvement of much greater numbers than were witnessed here.

Our results indicate key conditions during which the northwards migration of hoverflies on the West Coast of the USA can become observable. Two factors appear to be particularly important in this case, the first is the presence of a headland, constrained on one side by the Pacific Ocean, and on the other by the mountains of Montaña de Oro State Park (figure 1*b*). Hence, as in previous observations in the USA, Scandinavia and the UK, it seems that the coastal topology served to concentrate migrant numbers [6,7]. The second factor is the presence of a headwind, which is known to drive migrating hoverflies closer to the ground to reduce flight costs [8]. Given this, it is interesting to speculate why this event had not been reported before. Conditions in 2017 were good for other migrants such as the painted lady butterfly *Vanessa cardui* [14] and these same conditions may have also led to a boom in hoverfly populations. For example, during the first part of 2017, Central Coast California saw the highest rainfall since 1998 and warm spring temperatures (electronic supplementary material, file S5). Warmer spring temperatures have been shown to increase reproduction in overwintering insects and enhance the development and survival of their offspring [15]. By contrast, 2018 was a poorer year with a late spring and we failed to observe migration in the same area, albeit only over a two-week window of observation in April. We suspect that the lack of observations is also due to the relatively small size of hoverflies, a wide range of annual variation in numbers and the scarcity of geographically suitable sites for observations. In addition, the majority of migratory movement may actually occur at high altitude [16], restricting ground level observations to periods when conditions become unfavourable higher up. In support of this, 18 April 2017 saw over 10 inches of rain in San Luis Obispo County, followed by increasing temperatures and wind speeds leading up to the migration event 2 days later (electronic supplementary material, file S5). These meteorological conditions may have served to delay and concentrate numbers, or

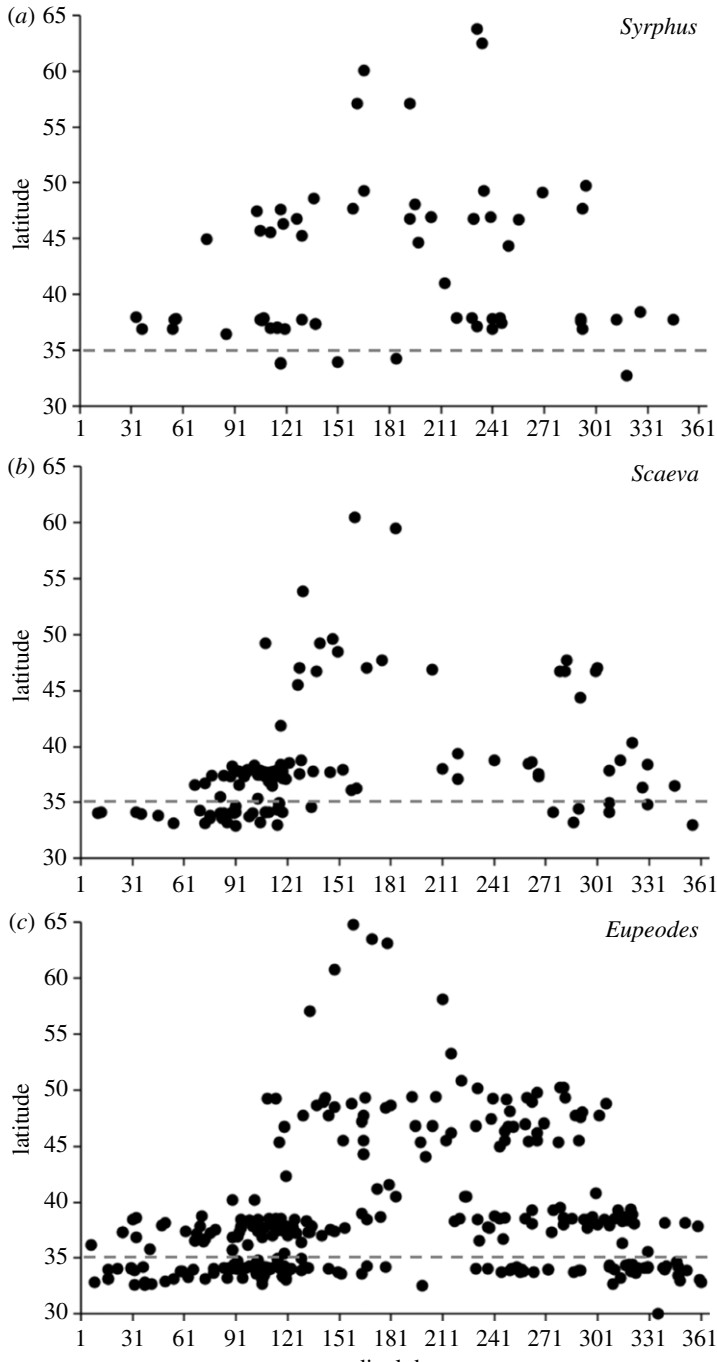

**Figure 3.** Candidate migrants and their observed seasonal distributions. Recordings of candidate hoverfly species from iNaturalist are plotted against latitude and ordinal date of observation. The dashed line marks the latitude of our observation at Montaña de Oro State Park. Records were taken west of −144° longitude and include all years 2007–2018. (a) Syrphus, (b) Scaeva and (c) Eupeodes include all species recorded from the respective genera, due to the difficulties associated with separating species in the field. Latitudes: California 32–42°, Oregon 42–46°, Washington 45–49°, British Columbia 48–60, Alaska 51–71°.

alternatively, may have made conditions aloft unfavourable. A final contributing factor may be the lack of public awareness of hoverflies as migrants, meaning that even when migration is observed, it may go unreported or be mistaken for another species. For example, huge influxes of hoverflies have previously caused panic as they were thought to be swarms of wasps [17].

Migration is widespread and well documented within European hoverflies [8,10], and many of these species or their congeners are also present in North America. Of those identified by us in California during the spring of 2018, we found seasonal patterns of abundance for *Scavea*, *Syrphus* and *Eupeodes* spp. from analysis of citizen science data (figure 3). Citizen science has previously been

used to infer migration in a range of insects based on the latitude and number of observations [18–20]. We identified a pattern of seasonal presence in our candidate hoverflies in southern California during spring, followed by Oregon, Washington, British Columbia and into Alaska through the summer, then the reverse into California during the autumn. These movements of *Scavea*, *Syrphus* and *Eupeodes* spp. are highly indicative of migratory behaviour. In support of our results, we note that *Eupeodes* spp. have previously been reported to spike in numbers in March in the Los Angeles Basin, then disappear from the region over summer, which again is indicative of an influx of migrants, although no directional movements have been noted [21]. Unfortunately, little evidence currently exists connecting these observations with active migratory behaviour. In the case of the 2017 migratory event, we tentatively believe, given the size, colour and ventral banding, that the most likely candidate is a species of *Syrphus*, although this requires confirmation through the capture of active migrants. However, our evidence does not rule out the presence of multiple species, a scenario that may be likely given the diversity of hoverflies found in North America, the prevalence of migratory life histories within the family and the mix of species often found during migrations in Europe [8,10].

## 5. Conclusion

Our study demonstrates the presence of migratory life histories of some hoverflies on the West Coast of North America and quantifies the scale of a huge migratory event that occurred in 2017 where hundreds of thousands of hoverflies passed over a 200 m front in only 30 min. The only previous reports of hoverfly migration within the USA occurred over 90 years ago on the East Coast [12], hence there remains a considerable gap in our knowledge of how alterations in land use, agriculture and climate may be affecting these events. The challenge now is to begin to document these migrations and their distribution without the obvious barriers that serve to concentrate migrants in Europe. The recruitment of citizen scientists and efforts to raise awareness of hoverfly migration are obvious ways to extend the search, as is the use of directional trapping stations at known funnelling points (e.g. [22]), all of which could be achieved with only a modest investment. Greater effort needs to be put into understanding the movement ecology of these insects, given the global context of pollinator declines [23] and the poleward spread of crop pests [24]. Therefore, we wish to issue a call to intensify the study and the search for hoverfly migration in the USA and elsewhere.

Ethics. Permission to collect hoverflies was obtained from the State of California Natural Resources Agency: Department of Parks and Recreation (Ref: 17-820-38).

Data accessibility. The datasets supporting this article have been uploaded as part of the electronic supplementary material. A full resolution version of electronic supplementary material, file S1 can be found here: https://doi.org/10.6084/m9.figshare.7584386.v1.

Authors' contributions. Hoverfly migration was observed and recorded by B.V.B. Citizen science data collation and analysis were carried out by K.R.W. and M.H.M.M. All authors contributed to sample collection and manuscript writing.

Competing interests. We have no competing interests.

Funding. Funding to K.R.W. was provided by the Royal Society through a University Research Fellowship (UF150126). M.H.M.M. received funding from the European Union's Horizon 2020 research and innovation programme under the Marie Skłodowska-Curie Grant Agreement no. 795568 (InsectMigration).

Acknowledgements. We thank Vincent Cicero from California Department of Fish & Game and Rouvaishyana from California State Parks for assistance obtaining permission to study flies in Montaña de Oro State Park, and Freddy Otte of the city of San Luis Obispo for permission to conduct research at Laguna Lake park. We also thank Giar-Ann Kung for logistical assistance, Jim and Celeste Royer for hospitality, Wolfgang Nentwig for supporting the study and all contributors to the iNaturalist platform.

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
