## [Reviewer comments · Royal Society Open Science]

Review History

RSOS-190153.R0 (Original submission)

Review form: Reviewer 1

Is the manuscript scientifically sound in its present form?

Yes

Are the interpretations and conclusions justified by the results?

Yes

Is the language acceptable?

Yes

Is it clear how to access all supporting data?

Yes

Do you have any ethical concerns with this paper?

No

Have you any concerns about statistical analyses in this paper?

No

Recommendation?

Accept with minor revision (please list in comments)

Comments to the Author(s)

The authors document an apparent hoverfly migration along the central California coast. This would be the first such migration reported for the western US and thus important to include in the literature. The paper is overall very well written. Below are a few minor issues and a suggestion for an improvement.

Page 4, final line: you probably meant the West Coast, not East Coast.

Page 8, line 40: This sentence is awkwardly worded and on line 47 "where" should be "were". It would be helpful to state here that observed European hoverfly migrations include multiple species, if that is true.

High numbers observed in 2017 could be due to good overwintering conditions, as described, and it could also be related to specific weather patterns. It would be nice to compare weather conditions on the days before the observed 2017 migration to those in 2018. Is there any evidence of poor conditions at high altitudes that might have brought the hoverflies to ground? For example, frontal passage or storms in the preceding day or two could have altered wind direction aloft, and without those conditions in 2018 similar numbers would not have been observed at ground level. Alternatively, storm conditions may have delayed and concentrated movement leading to the 13 April observation. Historical weather data are available for specific locations from Wunderground, and from NOAA.

Review form: Reviewer 2

Is the manuscript scientifically sound in its present form?

Yes

Are the interpretations and conclusions justified by the results?

Yes

Is the language acceptable?

Yes

Is it clear how to access all supporting data?

Not Applicable

Do you have any ethical concerns with this paper?

No

Have you any concerns about statistical analyses in this paper?

No

Recommendation?

Accept as is

Comments to the Author(s)

The authors have presented a quantitative method to understand hoverfly movement ecology. It is an important update in this research area.

Decision letter (RSOS-190153.R0)

07-Mar-2019

Dear Dr Wotton

On behalf of the Editors, I am pleased to inform you that your Manuscript RSOS-190153 entitled "Quantification of migrant hoverfly movements (Diptera: Syrphidae) on the West Coast of North America" has been accepted for publication in Royal Society Open Science subject to minor revision in accordance with the referee suggestions. Please find the referees' comments at the end of this email.

The reviewers and handling editors have recommended publication, but also suggest some minor revisions to your manuscript. Therefore, I invite you to respond to the comments and revise your manuscript.

- Ethics statement

- Data accessibility

<http://datadryad.org/submit?journalID=RSOS&manu=RSOS-190153>

- Competing interests

- Authors' contributions

All submissions, other than those with a single author, must include an Authors' Contributions section which individually lists the specific contribution of each author. The list of Authors should meet all of the following criteria; 1) substantial contributions to conception and design, or

acquisition of data, or analysis and interpretation of data; 2) drafting the article or revising it critically for important intellectual content; and 3) final approval of the version to be published.

- Acknowledgements

- Funding statement

Because the schedule for publication is very tight, it is a condition of publication that you submit the revised version of your manuscript before 16-Mar-2019. Please note that the revision deadline will expire at 00.00am on this date. If you do not think you will be able to meet this date please let me know immediately.

- 1) A text file of the manuscript (tex, txt, rtf, docx or doc), references, tables (including captions) and figure captions. Do not upload a PDF as your "Main Document";

- 2) A separate electronic file of each figure (EPS or print-quality PDF preferred (either format should be produced directly from original creation package), or original software format);
- 3) Included a 100 word media summary of your paper when requested at submission. Please ensure you have entered correct contact details (email, institution and telephone) in your user account;
- 4) Included the raw data to support the claims made in your paper. You can either include your data as electronic supplementary material or upload to a repository and include the relevant doi within your manuscript. Make sure it is clear in your data accessibility statement how the data can be accessed;
- 5) All supplementary materials accompanying an accepted article will be treated as in their final form. Note that the Royal Society will neither edit nor typeset supplementary material and it will be hosted as provided. Please ensure that the supplementary material includes the paper details where possible (authors, article title, journal name).

on behalf of Dr Emily Shepard (Associate Editor) and Kevin Padian (Subject Editor)
openscience@royalsociety.org

Associate Editor Comments to Author (Dr Emily Shepard):

Associate Editor: 1

Comments to the Author:

This short manuscript provides evidence for a novel hoverfly migration. Both reviewers are of the view that this represents an important insight to a data poor area. They provide a few minor suggestions to further improve the manuscript.

Reviewer comments to Author:

Reviewer: 1

Comments to the Author(s)

The authors document an apparent hoverfly migration along the central California coast. This would be the first such migration reported for the western US and thus important to include in the literature. The paper is overall very well written. Below are a few minor issues and a suggestion for an improvement.

Page 4, final line: you probably meant the West Coast, not East Coast.

Page 8, line 40: This sentence is awkwardly worded and on line 47 "where" should be "were". It would be helpful to state here that observed European hoverfly migrations include multiple species, if that is true.

High numbers observed in 2017 could be due to good overwintering conditions, as described, and it could also be related to specific weather patterns. It would be nice to compare weather conditions on the days before the observed 2017 migration to those in 2018. Is there any evidence of poor conditions at high altitudes that might have brought the hoverflies to ground? For example, frontal passage or storms in the preceding day or two could have altered wind direction aloft, and without those conditions in 2018 similar numbers would not have been observed at ground level. Alternatively, storm conditions may have delayed and concentrated movement leading to the 13 April observation. Historical weather data are available for specific locations from Wunderground, and from NOAA.

Reviewer: 2

Comments to the Author(s)

The authors have presented a quantitative method to understand hoverfly movement ecology. It is an important update in this research area.

Author's Response to Decision Letter for (RSOS-190153.R0)

See Appendix A.

Decision letter (RSOS-190153.R1)

12-Mar-2019

Dear Dr Wotton,

I am pleased to inform you that your manuscript entitled "Quantification of migrant hoverfly movements (Diptera: Syrphidae) on the West Coast of North America" is now accepted for publication in Royal Society Open Science.

on behalf of Dr Emily Shepard (Associate Editor) and Kevin Padian (Subject Editor)
openscience@royalsociety.org

Follow Royal Society Publishing on Twitter: [@RSocPublishing](https://twitter.com/RSocPublishing)
Follow Royal Society Publishing on Facebook:
<https://www.facebook.com/RoyalSocietyPublishing.FanPage/>
Read Royal Society Publishing's blog: <https://blogs.royalsociety.org/publishing/>

Appendix A

Centre for Ecology
and Conservation

Centre for Ecology and
Conservation
University of Exeter
Penryn Campus,
Penryn, Cornwall
TR10 9FE
UK
t: +44 (0)1326 25 4118
e: k.r.wotton@exeter.ac.uk
w: www.exeter.ac.uk

08 March 2019

Dear Dr Shepard,

We were pleased that both reviewers shared the view that our work represents an important insight into a data poor area. We have addressed the minor suggestions of reviewer 1 (see below) and I hope that you now find our manuscript ready for publication.

Sincerely yours,

Dr Karl R Wotton
Senior Lecturer & Royal Society University Research Fellow
University of Exeter, UK

Reviewer: 1 Comments and our responses (in red)

Page 4, final line: you probably meant the West Coast, not East Coast.

Correct. This has now been changed.

Page 8, line 40: This sentence is awkwardly worded and on line 47 “where” should be “were”. It would be helpful to state here that observed European hoverfly migrations include multiple species, if that is true.

We have changed this sentence to read:

“However, our evidence does not rule out the presence of multiple species, a scenario that may be likely given the diversity of hoverflies found in North America, the prevalence of migratory life histories within the family and the mix of species often found during migrations in Europe.”

High numbers observed in 2017 could be due to good overwintering conditions, as described, and it could also be related to specific weather patterns. It would be nice to compare weather conditions on the days before the observed 2017 migration to those in 2018. Is there any evidence of poor conditions at high altitudes that might have brought the hoverflies to ground? For example, frontal passage or storms in the preceding day or two could have altered wind direction aloft, and without those

conditions in 2018 similar numbers would not have been observed at ground level. Alternatively, storm conditions may have delayed and concentrated movement leading to the 20 April observation. Historical weather data are available for specific locations from Wunderground, and from NOAA.

We thank reviewer 1 for their suggestion. We have used Wunderground and the US Divisional Climate Data website (<https://wrcc.dri.edu/spi/divplot2map.html>) to check weather data spanning the 2017 migration event.

We have updated our text to include this information in the relevant locations:

“For example, during the first part of 2017, Central Coast California saw the highest rainfall since 1998 and warm spring temperatures (ESM file 5). Warmer spring temperatures have been shown to increase the reproduction of overwintering insects and to enhance the development and survival of their offspring (e.g. Ju et al. 2017 *Scientific reports*). In contrast, 2018 was a poorer year with a late spring and we failed to observe migration in the same area, albeit only over a two-week window of observation in April.”

and:

“In support of this, 18 April 2017 saw over 10 inches of rain in San Luis Obispo County, followed by increasing temperatures and wind speeds leading up the migration event two days later (ESM file 5). These meteorological conditions may have served to delay and concentrate numbers, or alternatively, may have made conditions aloft unfavourable.”